# Synthesis, Characterization and Visible-Light Photocatalytic Activity of Solid and TiO_2_-Supported Uranium Oxycompounds

**DOI:** 10.3390/nano11041036

**Published:** 2021-04-19

**Authors:** Mikhail Lyulyukin, Tikhon Filippov, Svetlana Cherepanova, Maria Solovyeva, Igor Prosvirin, Andrey Bukhtiyarov, Denis Kozlov, Dmitry Selishchev

**Affiliations:** 1Department of Unconventional Catalytic Processes, Boreskov Institute of Catalysis, 630090 Novosibirsk, Russia; lyulyukin@catalysis.ru (M.L.); smi@catalysis.ru (M.S.); kdv@catalysis.ru (D.K.); 2Schulich Faculty of Chemistry, Technion–Israel Institute of Technology, Haifa 32000, Israel; tihon@campus.technion.ac.il; 3Department of Catalysts Study, Boreskov Institute of Catalysis, 630090 Novosibirsk, Russia; svch@catalysis.ru; 4Department of Physicochemical Methods of Research, Boreskov Institute of Catalysis, 630090 Novosibirsk, Russia; prosvirin@catalysis.ru (I.P.); avb@catalysis.ru (A.B.)

**Keywords:** uranyl nitrate, titanium dioxide, photocatalysis, VOC oxidation, visible light, LED

## Abstract

In this study, various solid uranium oxycompounds and TiO_2_-supported materials based on nanocrystalline anatase TiO_2_ are synthesized using uranyl nitrate hexahydrate as a precursor. All uranium-contained samples are characterized using N_2_ adsorption, XRD, UV–vis, Raman, TEM, XPS and tested in the oxidation of a volatile organic compound under visible light of the blue region to find correlations between their physicochemical characteristics and photocatalytic activity. Both uranium oxycompounds and TiO_2_-supported materials are photocatalytically active and are able to completely oxidize gaseous organic compounds under visible light. If compared to the commercial visible-light TiO_2_ KRONOS^®^ vlp 7000 photocatalyst used as a benchmark, solid uranium oxycompounds exhibit lower or comparable photocatalytic activity under blue light. At the same time, uranium compounds contained uranyl ion with a uranium charge state of 6+, exhibiting much higher activity than other compounds with a lower charge state of uranium. Immobilization of uranyl ions on the surface of nanocrystalline anatase TiO_2_ allows for substantial increase in visible-light activity. The photonic efficiency of reaction over uranyl-grafted TiO_2_, 12.2%, is 17 times higher than the efficiency for commercial vlp 7000 photocatalyst. Uranyl-grafted TiO_2_ has the potential as a visible-light photocatalyst for special areas of application where there is no strict control for use of uranium compounds (e.g., in spaceships or submarines).

## 1. Introduction

Semiconductor photocatalysis has the potential for efficient utilization of light energy reaching the Earth from the Sun [1]. Commonly, semiconducting materials can be used for utilization of solar radiation via a number of routes: (i) to convert the energy of light directly into electricity by the photovoltaic effect using various silicon, cadmium telluride, copper indium gallium selenide, dye-sensitized (also well-known as Grätzel cells), perovskite, and other solar cells [2,3]; (ii) to store the energy in a form of chemical bond energy in fuels via photocatalytic or photoelectrochemical hydrogen evolution [4,5,6,7] and reduction of carbon dioxide [8,9,10]; (iii) to carry out useful chemical reactions, for instance, to synthesize high-value products via partial selective photocatalytic or photoelectrochemical oxidation [11,12] and to purify water or air via photocatalytic degradation and complete oxidation of pollutants [13,14]. Therefore, the photocatalytic processes can solve both energy and environmental issues.

Photocatalytic oxidation is one of main fields of photocatalysis and has the highest potential in the degradation of chemical and biological contaminants, which pollute air, water, and surfaces [15,16,17]. Wide-bandgap metal oxide semiconductors, namely TiO_2_, ZnO, WO_3_, exhibit high activities in this type of photocatalytic reaction and can be applied in a form of powder, thin film, or coating. TiO_2_ is the most commonly known and used photocatalyst for the degradation of air and water pollutants due to its chemical stability, simple preparation from minerals and other precursors, low cost, and high activity [18,19,20,21,22]. However, the action spectrum of TiO_2_ is restricted to the UV region due to its wide band gap that commonly has a value of 3.0–3.2 eV [23,24,25]. According to ASTM G-173-03 [26], a fraction of UV light in the solar radiation does not exceed 5%. Therefore, conventional TiO_2_ photocatalysts have a high performance only under radiation from artificial UV-light sources (e.g., mercury lamps, UV light-emitting diodes). Search and development of new materials is an important task for efficient utilization of solar radiation.

Current status and challenges in the development of materials for photocatalytic oxidation under visible light can be found in the recently published review [17]. Despite the other discovered materials (e.g., bismuth oxyhalides, graphitic carbon nitride), which can absorb visible light and have photocatalytic activity in this region, the modification of TiO_2_ to extend its action spectrum to the visible region still remains the main approach to develop an efficient photocatalyst for visible light. Sensitization with dyes, doping with metals or non-metals, metal loading, combination with narrow-bandgap semiconductors are possible methods for modification of TiO_2_ that provide its activation under visible light [17,27]. In addition to these methods, we have previously shown that the surface modification of TiO_2_ with uranyl ion (${\mathrm{UO}}_{2}^{2+}$) results in a photocatalyst that has great activity under visible light and can completely oxidize vapor of organic compounds at a high rate [28,29,30]. The photocatalytic activity of this system under visible light is due to the excitation of uranyl ion and reverse redox transformations between U^6+^ and U^4+^ charge states during the interactions with organic compounds and oxygen.

Many studies describe the photochemical and photocatalytic properties of uranyl ion itself in homogeneous solutions [31,32,33,34,35] or in heterogenous systems where it is grafted on the surface of porous supports [36,37,38,39,40]. Less information can be found on visible-light photocatalytic activity of other uranium compounds, especially solid materials, which have a benefit from the practical point of view compared to homogeneous systems. Uranium oxides are known to be semiconductors with a band gap less than 3 eV [41] and can potentially have photocatalytic activity in the oxidation of organic pollutants under visible light. It was our motivation to perform a mechanistic study for examining photocatalytic activity of various solid and supported uranium compounds.

In this study, we synthesize several uranium oxycompounds using a UO_2_(NO_3_)_2_ precursor, characterize them using physical methods, and evaluate their photocatalytic activity in a test reaction of acetone oxidation under blue light in a continuous-flow setup. A correlation between physicochemical properties of materials and their photocatalytic activities is discussed. The values of photonic efficiency are shown for all uranium samples, as well as for the commercial visible-light-active photocatalyst TiO_2_ KRONOS^®^ vlp 7000, tested under the same conditions, to give other researchers references for comparison of photocatalytic activity.

## 2. Materials and Methods

### 2.1. Materials

Uranyl nitrate hexahydrate (UO_2_(NO_3_)_2_·6H_2_O) from Isotope JSC (Moscow, Russia) was used as a uranium-containing precursor for the synthesis of uranium compounds and for the modification of a TiO_2_ photocatalyst. High purity or reagent grade chemicals, namely ethylenediamine (NH_2_CH_2_CH_2_NH_2_, 99.5%), ethanol (C_2_H_5_OH, 99.5%), ammonium hydroxide solution (NH_4_OH, 28%), sodium borohydride (NaBH_4_, 98%), and hydrazine hydrate (N_2_H_4_·xH_2_O, 64%) were used during the synthesis of samples as received from Sigma-Aldrich Corp. (St. Louis, MO, USA) without further purification. Commercial UV-active photocatalyst TiO_2_ Hombifine N from Sachtleben Chemie GmbH (Duisburg, Germany) was modified with uranyl nitrate to provide photocatalytic activity under visible light [29,30]. Additionally, commercial photocatalyst TiO_2_ KRONOS^®^ vlp 7000 (95% anatase, a_s,BET_ = 250 m^2^ g^–1^) from KRONOS TITAN GmbH (Leverkusen, Germany), which has activity both under UV and visible light, was used as a reference sample for benchmarking. High purity grade acetone (CH_3_COCH_3_, 99.5%) from AO REAHIM Inc. (Moscow, Russia) was used as a test organic compound for oxidation during the photocatalytic experiments.

### 2.2. Characterization Techniques 

The content of uranium in the samples was measured by X-ray fluorescence (XRF) analysis using an ARL ADVANT’X device (Thermo Fisher Scientific Inc., Waltham, MA, USA) equipped with a Rh anode of the X-ray tube. The phase composition was analyzed by X-ray powder diffraction (XRD) using a D8 Advance diffractometer (Bruker, Billerica, MA, USA) under CuK_α_ radiation. The scanning procedure was performed using a linear detector in the 2θ range of 5–90° with a step of 0.05°. Thermal analysis was performed in the temperature range of 20–1200 °C using an STA 449C instrument from NETZSCH-Gerätebau GmbH (Selb, Germany). The heating rate during the analysis was 10 °C g^–1^. The textural properties were investigated by N_2_ adsorption at 77 K using an ASAP 2400 instrument (Micromeritics Instrument Corp., Norcross, GA, USA). The surface area was calculated by BET analysis of N_2_ adsorption/desorption isotherms. The morphology was investigated by transmission electron microscopy (TEM) using a JEOL-2010 microscope (JEOL Ltd., Tokyo, Japan). TEM micrographs were received at an accelerating voltage of 200 kV and a resolution of 0.14 nm. The surface composition was investigated by X-ray photoelectron spectroscopy (XPS) using a SPECS photoelectron spectrometer (SPECS Surface Nano Analysis GmbH, Berlin, Germany) equipped with a PHOIBOS-150 hemispherical energy analyzer and an AlK_α_ radiation source (*hν* = 1486.6 eV, 150 W). The binding energy (BE) scale was pre-calibrated using the positions of Au4f_7/2_ (84.0 eV) and Cu2p_3/2_ (932.67 eV) photoelectron lines from metallic gold and copper foils. The background pressure at the analysis chamber did not exceed 8 × 10^–7^ Pa. The detailed Ti2p, C1s, and U4f spectral regions were recorded with a step of 0.1 eV. The C1s peak at 284.8 eV, attributed to carbon impurities, was used as an internal standard for the calibration of spectra. Experimental spectra were fitted using an approximation function based on a combination of the Gaussian and Lorentzian functions with subtraction of a Shirley-type background [42]. The data processing and peak fitting were performed using XPSPeak 4.1 software. The ratio between charge states of uranium was calculated using the areas of corresponding fitted peaks. The optical properties were investigated by UV–vis spectroscopy. The diffuse reflectance spectra (DRS) were recorded at room temperature in the range of 250–850 nm with a resolution of 1 nm using a Cary 300 UV–vis spectrophotometer (Agilent Technologies Inc., Santa Clara, CA, USA) equipped with a DRA-30I diffuse reflectance accessory. Special pre-packed polytetrafluoroethylene (PTFE) from Agilent was used as the reflectance standard. FT-Raman spectra were collected using a Bruker RFS 100/S spectrometer at 180° geometry in the range of 3700−100 cm^−1^ with a resolution of 4 cm^−1^. Excitation of the 1064-nm line was provided by a Nd-YAG laser with output power of 100 mW.

### 2.3. Photocatalytic Experiments

A continuous-flow setup was used for the evaluation of photocatalytic activity of synthesized and reference samples in a test reaction of acetone oxidation under blue light [43]. Each sample was uniformly deposited on a round glass plate with an area of 9.1 cm^2^ to obtain a layer with an area density of 2 mg cm^−2^ and was placed into the photoreactor. The temperature of photoreactor was 40.0 ± 0.1 °C. The mass flow controllers were adjusted to obtain the inlet humidified air with relative humidity of 20 ± 1% and volume flow rate of 0.069 ± 0.001 L min^−1^. The plate with sample was irradiated using a 100-W light-emitting diode (LED) with a maximum of emission at 450 nm (see Appendix A for details). The specific total irradiance of the sample measured using an ILT 950 spectroradiometer from International Light Technologies Inc. (Peabody, MA, USA) was 93.5 mW cm^−2^. Acetone was selected as a test organic compound for oxidation, and its inlet concentration in the gas phase was 29 ± 2 μmol L^−1^. The inlet and outlet gas mixtures were periodically analyzed using an FT-IR spectrometer FT-801 from Simex LLC (Novosibirsk, Russia) equipped with an IR long-path gas cell from Infrared Analysis Inc. (Anaheim, CA, USA). The quantitative analysis was performed based on the Beer–Lambert law via the integration of collected IR spectra in the range of 1160–1265 cm^−1^ for the evaluation of acetone concentration and of 2200–2400 cm^−1^ for the evaluation of CO_2_ concentration. The former range (i.e., 1160–1265 cm^−1^) is attributed to the absorption band due to vibration of C–C bond ($\nu_{C–C\;}$) in acetone molecule, and the latter is attributed to the absorption bands due to vibrations in CO_2_ molecule. The corresponding attenuation coefficients for acetone and CO_2_ were estimated from the preliminary calibration data.

Steady-state rate of CO_2_ formation during the acetone oxidation, evaluated as the product of the volume flow rate and the difference between outlet and inlet concentrations of CO_2_, was used as the photocatalytic activity of the samples. The photonic efficiency of CO_2_ formation ($\xi_{CO_{2}})$ was estimated to evaluate the efficiency of light utilization as follows:$$\xi_{CO_{2}}=\frac{W_{CO_{2}}}{q_{n,p}^{0}}\times 100\%$$
where $W_{CO_{2}}$ is the measured steady-state rate of CO_2_ formation (μmol min^–1^), $q_{n,p}^{0}$ is the total incident photon flux ($q_{n,p}^{0}$ = 32.4 μmol min^–1^).

## 3. Results and Discussion

The photocatalytic activity was expected to strongly depend on the charge state of uranium and the composition of its compound. A lot of effort in this study was spent on the synthesis and characterization of different uranium compounds. The results are shown below sequentially from the synthesis and characterization of samples to evaluation of their photocatalytic activity and discussion of correlations with physicochemical characteristics.

### 3.1. Synthesis and Characterization Data

One of the simple methods for the synthesis of uranium oxides is the thermal decomposition of uranyl nitrate that can lead to the formation of UO_3_ and U_3_O_8_ oxides. The thermogravimetric analysis of initial UO_2_(NO_3_)_2_·6H_2_O precursor was performed to select the temperature range for the preparation of corresponding oxide. The peaks in DTG and DTA curves at temperatures below 300 °C correspond to the loss of crystallization water and nitrate by UO_2_(NO_3_)_2_·6H_2_O (Figure 1). Uranium oxides form at higher temperatures. The formation of uranium(VI) oxide (i.e., UO_3_) starts at a temperature of 340 °C. A further increase in the calcination temperature above 570 °C leads to a loss of oxygen by UO_3_ and a partial reduction of uranium with the formation of a mixed U_3_O_8_ oxide. These results agree with the published data [44].

Based on the data described above, the temperatures of 400, 600, and 900 °C were selected to prepare uranium oxides. The calcination of uranyl nitrate at 400 °C was expected to result in the formation of UO_3_, whereas the temperatures of 600 and 900 °C were employed to obtain U_3_O_8_ with different textural characteristics.

According to XPS analysis (Figure 2a), the sample prepared via the calcination of UO_2_(NO_3_)_2_·6H_2_O at 400 °C contains uranium with the charge state of 6+. This result in combination with XRD data (Figure 2b) confirms the formation of UO_3_ oxide at this temperature. In addition to an amorphous form, UO_3_ commonly has seven polymorphic modifications from α to η [45]. Although the literature data mainly state that the calcination of uranyl nitrate in air at temperatures of 400–600 °C results in the formation of γ-UO_3_ [46], the XRD pattern of the sample prepared at 400 °C shows the best coincidence with β modification of UO_3_ (Figure 2b). The synthesized sample is referred in the paper as **UO_3_**. 

The content of uranium in the samples prepared via the calcination of uranyl nitrate at 600 and 900 °C is in the range of 86.4–87.8 wt.% which is close to the theoretically estimated value for the U_3_O_8_ compound (ω_U_ = 84.8 wt.%). XRD analysis also confirms that both samples have the crystal phase of U_3_O_8_ (Figure 2c). The sample prepared at 900 °C (i.e., **U_3_O_8_-T900**) exhibits a better separation of peaks in the XRD pattern and their higher intensity if compared with the sample prepared at 600 °C (i.e., **U_3_O_8_-T600**). This result indicates a higher crystallinity of the **U_3_O_8_-T900** sample and a larger size of its crystallites due to sintering. As has been shown in our previously published paper [29], the **U_3_O_8_-T900** sample contains uranium in the charge states of U^6+^ and U^4+^, and a ratio between these states is close to 2.

The hydrothermal method was employed in an attempt to synthesize a single uranium(IV) oxide (i.e., UO_2_). For this purpose, UO_2_(NO_3_)_2_·6H_2_O was dissolved in deionized water. Ethylenediamine was added to the solution until a 150-times molar excess compared to uranyl nitrate was reached. A Teflon lined autoclave was filled with the prepared solution, tightly closed, and stored in an electric oven at 160 °C for 72 h. The synthesized sample was thoroughly washed with deionized water and dried in air at 70 °C. The XRD pattern of this sample shows a very broad peak at ca. 17°, and narrow peaks, which correspond to the crystalline phase of UO_2_ (Figure 3a). There are no other broad peaks in the pattern, and the observed peak at ca. 17° can be attributed to an amorphous phase. It should be noticed that diffraction peaks attributed to the crystalline phase are shifted to high-angle region relative to the positions of UO_2_ peaks. This shift indicates modified parameters of crystal lattice in the prepared sample compared to the single crystal of UO_2_ possibly due to an excess of oxygen in interstitial positions [47,48]. Formation of nonstoichiometric, as well as mixed oxides is a common case for uranium [49,50]. The formula of the prepared sample can be expressed as UO_2+x_ with x value of 0.12 estimated from the XRD data (see Appendix A). It is important to note that this expression does not correspond to other stoichiometric uranium oxides, namely U_4_O_9_ (i.e., UO_2.25_) and U_3_O_7_ (i.e., UO_2.33_), because the estimated x value is much lower. The formula of UO_2.12_ gives a value of 4.2+ as a formal charge of uranium. The data of XPS analysis show the charge states of U^4+^ and U^5+^ for this sample which also confirms a nonstoichiometry (Figure 3c). The surface composition of the sample estimated using XPS gives a value of 4.8+ as a formal charge of uranium. This result indicates that the surface and bulk compositions of the prepared sample may substantially differ. An amorphous phase mentioned above for this sample may be a reason for that. Therefore, the sample prepared via the hydrothermal thermal treatment of uranyl nitrate with ethylenediamine is referred in the paper as **UO_2+x_**.

The hydrothermal treatment was also used for the preparation of a uranium compound with ammonia. For this purpose, uranyl nitrate was similarly autoclaved in an aqueous solution of ammonia and ethanol, which were both taken in a 10-times molar excess. Figure 3b shows the results of XRD analysis of the sample synthesized via this method. Except a small peak at 2θ = 13°, the XRD pattern of this sample shows a good coincidence with the phase of U_2_(NH_3_)O_6_·3H_2_O. This sample is referred in the paper as **U_2_(NH_3_)O_6_****·3H_2_O**.

XPS analysis of this sample shows the uranium charge states of U^5+^ and U^6+^ (Figure 3c). According to a ratio of peak areas, the surface content of U^5+^ and U^6+^ states are 34% and 66%, respectively. This ratio gives a value of 5.7+ as a formal charge of uranium, which is less than the value estimated from the formula of U_2_(NH_3_)O_6_·3H_2_O (i.e., 6+). An additional phase, which gives a small signal in the XRD pattern at 2θ = 13°, may be present in the sample and may contain uranium with the charge state of 5+ which reduces the total value of formal charge. A partial reduction of U^6+^ to U^5+^ may also occur in a spectrometer chamber under vacuum conditions and exposure to X-rays. The results of the physical methods illustrate that, in contrast to ethylenediamine, ammonia does not lead to a substantial reduction of uranium in uranyl ion under hydrothermal treatment and mainly affects the coordination sphere.

The third method used for the preparation of uranium compounds was the reduction of uranyl nitrate with sodium borohydride or hydrazine. These reducing agents were selected based on the values of standard redox potentials, which should be high enough for reduction of uranyl ion to uranium(IV) oxide. Concerning the results of physical analyses, it is difficult to make clear conclusions on the chemical and phase composition of the samples prepared via the chemical reduction of uranyl nitrate.

In the case of sodium borohydride, the crystal data of sodium polyuranates sufficiently fit the results of XRD analysis. The crystal phase of Na_2_U_7_O_22_ polyuranate shows the best coincidence with the XRD pattern of the reduced sample (Figure 4a). A broad peak in the range of 22–30° in the XRD pattern indicates an amorphous phase in the sample. A faint but distinct peak at 1071.6 eV in the photoelectron Na1s spectral region confirms the presence of sodium in the prepared sample (see the inset in Figure 4c). The XPS data also show U^5+^ and U^6+^ charge states with the surface contents of 31 and 69%, respectively (Figure 4c). On the other hand, a mass fraction of uranium in the prepared sample measured using XRF analysis is 71.2 wt.%. This value does not correspond to the value theoretically estimated for sodium polyuranate of Na_2_U_7_O_22_ (i.e., ω_U_ = 82.6 wt.%), as well as for main uranium oxides (88.2, 84.8, and 83.2 wt.% for UO_2_, U_3_O_8_, and UO_3_, respectively). Sodium diuranate (i.e., Na_2_U_2_O_7_) with ω_U_ = 75.1 wt.% more correlates to the measured mass fraction of uranium. This indicates that the chemical composition of the prepared sample may differ from Na_2_U_7_O_22_ polyuranate, which is expected from XRD analysis. An amorphous phase in the sample may also be a reason for the reduced value of uranium content. Therefore, this sample is referred in the paper as **NaU_x_O_y_**.

In the case of hydrazine, XPS analysis (Figure 4c) shows only the U^6+^ charge state of uranium in the prepared sample. XRF analysis gives a value of 80.1 wt.% as a mass fraction of uranium, and this value is lower than the fraction corresponding to the stoichiometric UO_3_ oxide (see above). Actually, the XRD pattern of the sample prepared using hydrazine does not correspond to the crystal data of the UO_3_ oxide while the data for crystalline UO_3_·H_2_O hydrate (or UO_2_(OH)_2_) shows the best coincidence (Figure 4b). Therefore, this sample is referred in the paper as **UO_3_·H_2_O**. On the contrary to our expectations, both sodium borohydride and hydrazine as reducing agents do not provide a strong reduction of uranium during interaction with uranyl nitrate.

Along with solid uranium oxycompounds, we investigate supported materials based on nanocrystalline TiO_2_. For this purpose, a superfine anatase TiO_2_ was impregnated with aqueous solution of uranyl nitrate followed by drying at 160 °C. A mass loading of UO_2_(NO_3_)_2_ in the sample was 5%. Detailed information on the synthesis can be found in our previously published paper [29]. This sample is referred in the paper as **UO_2_(NO_3_)_2_/TiO_2_**.

The modification of TiO_2_ with uranyl nitrate changes its optical properties and results in appearance of light absorption in the range of 400–530 nm (Figure 5a). This absorption corresponds to the spectrum of uranyl nitrate with differences as follows: (i) fine structure of the absorption is not observed in the supported sample; (ii) maximum of absorption is shifted to the long-wave region. Other physical methods indicate a chemisorbed state of uranyl ion on the TiO_2_ surface. According to TEM and XRD analyses (Figure 5b,c), no crystalline particles of UO_2_(NO_3_)_2_ are present on the TiO_2_ surface. An averaged size of TiO_2_ crystallites, 9 nm, does not change after surface modification with uranyl nitrate.

Figure 6 shows Raman spectra. The peaks at 150 cm^–1^ (E_g_), 397 cm^–1^ (B_1g_), 514 cm^–1^ (B_1g_ + A_1g_), and 638 cm^–1^ (E_g_), that are specific for anatase, are observed for pristine and modified TiO_2_ samples. As shown in literature [51], the Raman peak at 150 cm^−1^ is extremely sensitive to the impurities of uranium into the crystal lattice of anatase because these impurities change the shape of the peak and position of its maximum. In our case, the shape and position are similar which indicates the absence of new impurities in the crystal lattice after modification. A slight decrease in the intensity of the signal for the **UO_2_(NO_3_)_2_/TiO_2_** sample compared to the pristine TiO_2_ is due to surface shielding by uranyl ions.

Absorption bands at 832 and 1049 cm^–1^ for the uranyl-modified TiO_2_ sample (Figure 6b) correspond to symmetric valence oscillations of uranyl ion and the nitrate group, respectively [52,53]. An increase in the loading of uranyl nitrate from 5 to 10 wt.% results in double increase in the intensity of these bands (Figure 6c). The positions of the bands substantially differ from the signals for initial crystalline uranyl nitrate (Figure 6a). The absence of signals at 874 and 1038 cm^–1^, attributed to crystalline uranyl nitrate, in the spectra of uranyl-modified samples (Figure 6b,c), indicates a chemisorbed state of uranyl ions on the TiO_2_ surface. The position of the signal for the nitrate group at 1049 cm^–1^, which is commonly attributed to isolated groups on the TiO_2_ surface [54], also supports this conclusion.

XPS analysis shows the U^6+^ charge state in uranyl ion chemisorbed on the TiO_2_ surface (Figure 7a). It is important to note that TiO_2_ as a support for uranyl ion promotes fast and reversible transformations between U^6+^ and U^4+^ charge states [29]. Under vacuum conditions and exposure to X-rays, a partial reduction of U^6+^ to U^4+^ occurs but oxygen treatment results in a fast reoxidation back to U^6+^.

Hydrothermal treatment of **UO_2_(NO_3_)_2_/TiO_2_** in water solution with ethanol at 160 °C for 48 h (i.e., **UO_2_(NO_3_)_2_/TiO_2_-HT**) and thermal treatment of **UO_2_(NO_3_)_2_/TiO_2_** in air at 500 °C for 3 h (i.e., **UO_2_(NO_3_)_2_/TiO_2_-T500**) were employed to change the chemical state of uranyl ion on the TiO_2_ surface. No uranium-contained crystal phases are detected in the samples after these treatments using XRD analysis. This result indicates that, similarly to the **UO_2_(NO_3_)_2_/TiO_2_** sample, uranium on the surface of treated samples presents in a form of chemisorbed species or small clusters and does not form large crystallites. XRD patterns of the treated samples show only peaks of anatase (Figure 7b). A substantial change in both cases is that the treated samples have higher size of crystallites, namely 9 nm for pristine TiO_2_ and 21–22 nm for the treated samples. These data correlate with the data on the specific surface area of initial **UO_2_(NO_3_)_2_/TiO_2_** and treated samples (Table 1). According to DRIFT analysis (Appendix A), both treatments result in the removal of nitrate groups from the surface of TiO_2_.

In contrast to **UO_2_(NO_3_)_2_/TiO_2_**, XPS analysis shows three charge states of uranium for the autoclaved sample while only the U^4+^ charge state for the sample calcined in air. We expected that the calcination of **UO_2_(NO_3_)_2_/TiO_2_** at 500 °C would result in the formation of solid uranium oxide on the surface of TiO_2_ but as mentioned above, no corresponding peaks are observed in the XRD pattern. The treatment at a higher temperature was not employed to avoid the phase transformation of TiO_2_ from anatase to rutile.

As a result, we synthesize and characterize the main uranium oxides (i.e., UO_3_, U_3_O_8_, UO_2_), other uranium compounds (U_2_(NH_3_)O_6_·3H_2_O, polyuranate, UO_3_·H_2_O), and TiO_2_-supported materials contained uranium in a form of chemisorbed uranyl ions and species. Table 1 summarizes the names of all prepared samples, short description of their synthesis, and physicochemical characteristics. The next section describes the photocatalytic activity of prepared materials under blue light.

### 3.2. Photocatalytic Activity

All the synthesized uranium compounds and TiO_2_-supported materials have a photocatalytic activity and are able to completely oxidize acetone vapor under blue light (450 nm, see Appendix A). It indicates that the potentials of excited states or charge carriers photogenerated under blue light are high enough that they can be involved in redox interactions with organic molecules and oxygen. The photocatalytic oxidation of acetone to CO_2_ and water occurs over these samples without the formation of gaseous intermediates. The figures below show the data on steady-state rate and photonic efficiency of CO_2_ formation during the oxidation of acetone. The results are divided on the groups of solid uranium oxycompounds and TiO_2_-supported materials. As absorption of light is an essential step for photocatalytic reactions, UV–vis spectra of concerning samples are also shown in figures and discussed. Commercial photocatalyst TiO_2_ KRONOS^®^ vlp 7000, which has activity both under UV and visible light, plays in this study a role of benchmark to give other researchers possibility for comparison of the results.

#### 3.2.1. Solid Uranium Oxycompounds

Figure 8 shows the data on photocatalytic activity of the prepared solid uranium oxycompounds and their UV–vis diffuse reflectance spectra. Uranyl nitrate is known to have a series of absorption bands between 360 and 500 nm, which form a broad absorption peak in this region (Figure 8b). Pristine uranyl nitrate has photocatalytic activity under blue light (i.e., 450 nm). The steady-state rate of CO_2_ formation during the oxidation of acetone over solid uranyl nitrate is 0.021 μmol min^–1^ which corresponds to the photonic efficiency of 6.5 × 10^–2^%. These values are only 2 times lower than the corresponding values for the visible-light photocatalyst TiO_2_ KRONOS^®^ vlp 7000: 0.043 μmol min^–1^ and 13.3 × 10^–2^%. Despite an intense absorption of light in the corresponding region, uranium oxides prepared via the calcination method (i.e, **UO_3_** and **U_3_O_8_** samples) exhibit very low photocatalytic activity, (2.4–4.2) × 10^–3^ μmol min^–1^, which is only slightly higher than the detection limit for the used experimental setup (1.5 × 10^–3^ μmol min^–1^). A low surface area of the synthesized samples may be a reason for that because this parameter commonly affects the activity of heterogenous photocatalysts. As a support for this statement, an increase in the calcination temperature of uranyl nitrate from 600 to 900 °C leads to a decrease in the surface area of U_3_O_8_ oxide from 5.3 to 0.6 m^2^ g^–1^ (Table 1) and simultaneous decrease in its activity about 2 times, despite an enhanced crystallinity of the sample calcined at higher temperature (Figure 2c). On the contrary to our expectations, the **UO_2+x_** sample prepared via the hydrothermal treatment of uranyl nitrate in water solution with ethylenediamine exhibits an extremely low surface area but the photocatalytic activity of this sample (i.e., 3.3 × 10^–3^ μmol min^–1^) is comparable to the activity of UO_3_ and U_3_O_8_ oxides. These results indicate that an extended surface area is not the only crucial parameter for the activity. Electrical conductivity of materials can also play an important role. According to previous works [55,56,57,58], electrical conductivity of uranium oxides increases similarly to the mass fraction of uranium in a sequence from UO_3_ to UO_2_. Additionally, hyperstoichiometric UO_2_ oxide is a p-type semiconductor while the UO_3_ and U_3_O_8_ oxides are n-type semiconductors [59]. Small values of observed photocatalytic activity of uranium compounds do not allow us to make a clear conclusion on the key parameter that affects the photocatalytic activity.

Other prepared uranium oxycompounds, which contain uranium in the charge state close to 6+, exhibit much higher photocatalytic activity. The **U_2_(NH_3_)O_6_·3H_2_O** sample prepared via the hydrothermal treatment of uranyl nitrate in water solution with ethanol and ammonia exhibits the largest specific surface area among all the prepared solid uranium oxycompounds (i.e., 19.1 m^2^ g^–1^, Table 1). The photocatalytic activity of this sample is 0.011 μmol min^–1^ which is 2 times lower than the activity of pristine uranyl nitrate. It is important to note that ammonia is known to undergo photocatalytic oxidation as well as organic compounds. Therefore, NH_3_-groups in the composition of the **U_2_(NH_3_)O_6_·3H_2_O** sample may interact with photogenerated holes, suppressing the pathway of their interaction with acetone and water molecules, consequently reducing the activity of the sample in the target process and playing a negative role.

The **NaU_x_O_y_** sample, which is prepared via the chemical reduction of uranyl nitrate with sodium borohydride and has crystal phase of a polyuranate, exhibits substantially higher photocatalytic activity (0.018 μmol min^–1^), despite a low value of specific surface area (Table 1). It is further proof that not only textural properties strongly affect the photocatalytic activity. The highest activity among all the synthesized solid uranium oxycompounds is detected in the case of the **UO_3_·H_2_O** sample. The rate and photonic efficiency of CO_2_ formation for this sample (0.065 μmol min^–1^ and 0.2%) are 1.5 times higher than the corresponding values for visible-light photocatalyst TiO_2_ KRONOS^®^ vlp 7000.

Therefore, the chemical and phase compositions play a key role in the photocatalytic activity. Except the uranium(VI) oxide prepared using a high-temperature treatment, the uranium compounds, which contain uranium with the charge state close to 6+, exhibit much higher activity compared to other compounds with lower charge state. The UO_3_**·**H_2_O compound can be expressed as uranyl hydroxide (i.e., UO_2_(OH)_2_). A high activity of the **UO_3_·H_2_O** sample, as well as initial uranyl nitrate, additionally supports a specific role of uranyl ion, which provides reactive excited charge states under radiation. It is important to note that the presence of water or OH-groups in the crystal structure of the material, as in the case of the **UO_3_·H_2_O** sample, may be a reason for high activity of this sample due to an enhanced formation of hydroxyl radicals. The pathway through interactions with hydroxyl radicals commonly plays a key role in the oxidation of organic compounds in aqueous solutions and humidified air.

#### 3.2.2. TiO_2_-Supported Materials

Uranium-contained samples based on TiO_2_ photocatalyst exhibit much higher activity under blue light compared to the solid uranium oxycompounds. Figure 9 shows the corresponding values of steady-state rate and photonic efficiency of CO_2_ formation over TiO_2_-supported samples, initial precursors, and KRONOS^®^ vlp 7000 reference. UV–vis spectra of all materials are also shown in this figure.

TiO_2_ used as a support is 100% anatase with nanosized crystallites and has very high activity in the UV region corresponding to its bandgap absorption. Otherwise, its photocatalytic activity under blue light is extremely low (0.011 μmol min^–1^) and is mainly due to nitrogen impurities in its composition. In contrast to single TiO_2_ or solid uranyl nitrate, uranyl-grafted TiO_2_ (i.e., **UO_2_(NO_3_)_2_/TiO_2_**) exhibits the highest activity among all samples (0.73 μmol min^–1^ and 2.3%), which is 17 times higher than the activity of the commercial KRONOS^®^ vlp 7000 visible-light photocatalyst. It is important to note that the mentioned value of photonic efficiency shows the efficiency of product formation (i.e., CO_2_). Many researchers estimate the efficiency for the reaction in whole. This estimation is commonly based on an assumption that one photon absorbed by (or incident on) the photocatalyst changes the oxidation state of carbon and oxygen into the oxidizing compound and molecular oxygen by 1. Taking into account the stoichiometry of acetone oxidation, the value of 2.3% corresponds to 12.2% as the photonic efficiency of this reaction (see Appendix A for details). This is a very high value for the complete oxidation of organic compounds in the gas phase under visible light [60,61,62].

The photocatalytic activity of uranyl-contained materials under blue light is due to the excitation of uranyl ion and reverse redox transformations between U^6+^ and U^4+^ charges during the interactions with organic compounds and oxygen. If compared to the crystalline form of UO_2_(NO_3_)_2_, uranyl nitrate supported on the surface of TiO_2_ exhibits extremely higher photocatalytic activity. Two reasons may explain this synergistic effect: (i) TiO_2_, as well as other porous supports, increases the dispersion of uranyl nitrate and stabilizes a chemisorbed state of uranyl ion, as has shown above using characterization methods. This means that a higher number of U-sites are available for interactions with organic molecules and interfacial charge transfer under irradiation. This statement is supported by our previously published data [28] showing a linear dependence of activity on the content of uranyl nitrate up to 10 wt.%. (ii) The specific role of TiO_2_ as support for uranyl ions. As we have shown previously on comparison with silica and alumina supports [28,29], just TiO_2_ provides fast redox transformations between U^6+^ and U^4+^ charge states that results in a very high photocatalytic activity of this material.

As has shown above, the chemical reduction of uranyl nitrate with hydrazine leads to an increase in photocatalytic activity of the prepared sample (i.e., **UO_3_·H_2_O**) compared to initial uranyl nitrate (Figure 8a). In the case of uranyl-grafted TiO_2_, this treatment contrary decreases the photocatalytic activity. It is difficult to make clear conclusions on the transformations of uranyl ion occurred on the TiO_2_ surface during this treatment. Therefore, the data for other treatments, namely calcination and hydrothermal treatment, are discussed below to show an important role of uranyl ion in the concerning system. Both employed thermal treatments lead to substantial decrease in the activity of uranyl-grafted TiO_2_ (Figure 9a). The values of activity are 0.21 μmol min^–1^ (0.65%) and 0.10 μmol min^–1^ (0.30%) for hydrothermally treated (i.e., **UO_2_(NO_3_)_2_/TiO_2_-HT**) and thermally treated (i.e., **UO_2_(NO_3_)_2_/TiO_2_-T500**) samples, respectively. As mentioned in the previous section, no uranium-contained crystal phases are detected in the treated samples using XRD analysis. However, UV–vis spectra for these samples differ from the spectrum of prepared uranyl-grafted TiO_2_ sample because they exhibit much higher absorption of light in whole visible region (Figure 9b). A form of these spectra in the visible region corresponds to the spectra for solid U_3_O_8_ and UO_2_ oxides described above (Figure 8b), which have a strong absorption of light in whole visible region and are dark brown or black in color. Therefore, according to UV–vis spectroscopy, both treatments lead to a change in the chemical state of uranyl ion on the TiO_2_ surface which is also supported with the data of XPS analysis (Figure 7a). Most probably, complete or partial transformation of uranyl ions to clusters or highly dispersed oxide particles occurs on the surface during the treatments. It is important to note that if these TiO_2_-supported materials are compared with solid uranium oxycompounds prepared from uranyl nitrate via thermal and hydrothermal treatments, they exhibit much higher activity. The suggestion mentioned above on an increased dispersion of uranium compounds on the TiO_2_ surface is also relevant for this case.

The results of photocatalytic measurements indicate that only the state of uranyl ion with the charge of 6+ provides a high photocatalytic activity under blue light. Other states on the TiO_2_ surface with lower charge of uranium exhibit much lower activity. In the case of the **UO_2_(NO_3_)_2_/TiO_2_-HT** sample, an intermediate U^5+^ charge state (Figure 7a) interrupts a fast transition U^6+^↔U^4+^ and suppresses the photocatalytic activity. In the case of **UO_2_(NO_3_)_2_/TiO_2_-T500**, the equilibrium between uranium charge states is totally shifted to U^4+^ (Figure 7a), and the activity is limited by the number of uranium species able to change their oxidation state and participate in the transfer of the charge carrier for the occurring photocatalytic reaction. It should be noted that substantial decrease in the activity of treated samples can also be due to a strong decrease in the surface area of TiO_2_ support after these treatments (Table 1).

Therefore, this study shows that solid uranium oxycompounds have a low photocatalytic activity under blue light. Otherwise, uranium species immobilized on TiO_2_ support exhibit a very high activity. Uranyl-grafted TiO_2_ has the potential as visible-light photocatalyst for special areas of application where there is no strict control for use of uranium compounds (e.g., in spaceships or submarines). It also can be investigated in the reactions of photocatalytic reduction under visible light. 

## 4. Conclusions

We successfully synthesize and characterize the main uranium oxides (i.e., UO_3_, U_3_O_8_, UO_2_), other uranium compounds (U_2_(NH_3_)O_6_·3H_2_O, polyuranate, UO_3_·H_2_O), and TiO_2_-supported materials containing uranium in a state of chemisorbed uranyl ions and species. All the synthesized solid uranium oxycompounds and TiO_2_-supported materials have a photocatalytic activity in oxidation reactions and are able to completely oxidize acetone in the gas-phase under blue (450 nm) light. It confirms that the potentials of excited states or charge carriers photogenerated under blue light are high enough that they can be involved in redox interactions with organic molecules and oxygen. The data on steady-state rate of CO_2_ formation during the oxidation of acetone vapor under blue light show that the chemical and phase compositions play a key role in the photocatalytic activity of uranium compounds. Except the uranium(VI) oxide prepared using a high-temperature treatment, the uranium compounds, which contain uranium with the charge state of 6+, especially uranyl ion, exhibit much higher activity compared to other compounds with lower charge state. Among solid compounds, the highest activity is observed for the sample prepared via the chemical reduction of uranyl nitrate with hydrazine, which has the crystal structure of UO_3_·H_2_O hydrate (or UO_2_(OH)_2_).

Immobilization of uranyl ions on the surface of TiO_2_ nanocrystallites via the impregnation of TiO_2_ support with aqueous solution of uranyl nitrate results in a photocatalyst with great activity under blue light. The photonic efficiency of acetone oxidation over TiO_2_-supported material, 12.2%, is 17 times higher than the efficiency for commercial photocatalyst TiO_2_ KRONOS^®^ vlp 7000. Thermal and hydrothermal treatments of uranyl-grafted TiO_2_ decrease its photocatalytic activity due to a change in chemical state of uranium and a strong decrease in surface area of TiO_2_ support after these treatments.

## Figures and Tables

**Figure 1 nanomaterials-11-01036-f001:**
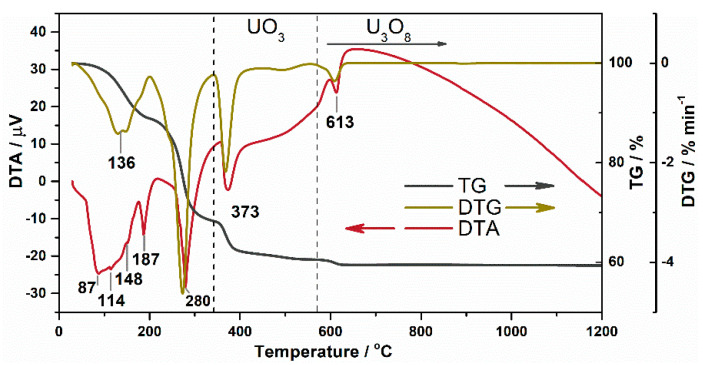
Thermogravimetric analysis of UO_2_(NO_3_)_2_·6H_2_O precursor.

**Figure 2 nanomaterials-11-01036-f002:**
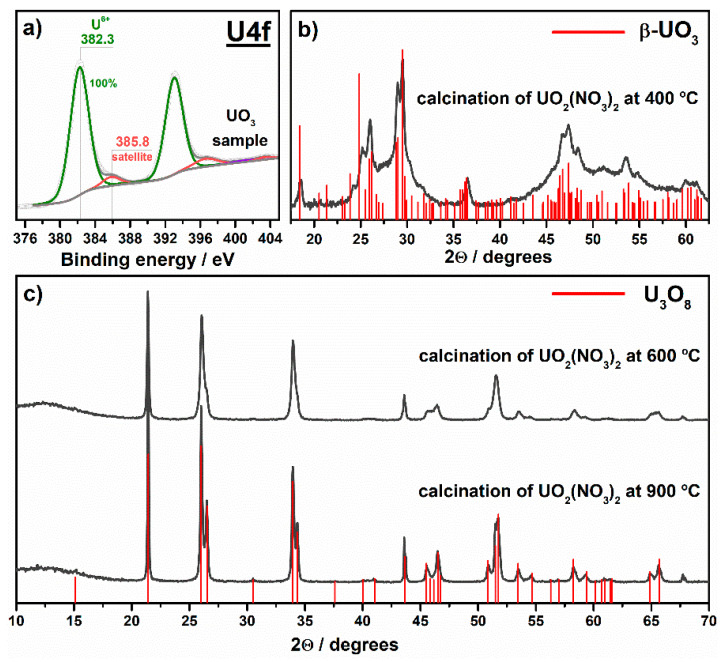
Photoelectron U4f spectral region (**a**) and XRD patterns (**b**,**c**) for the samples prepared via the calcination of uranyl nitrate.

**Figure 3 nanomaterials-11-01036-f003:**
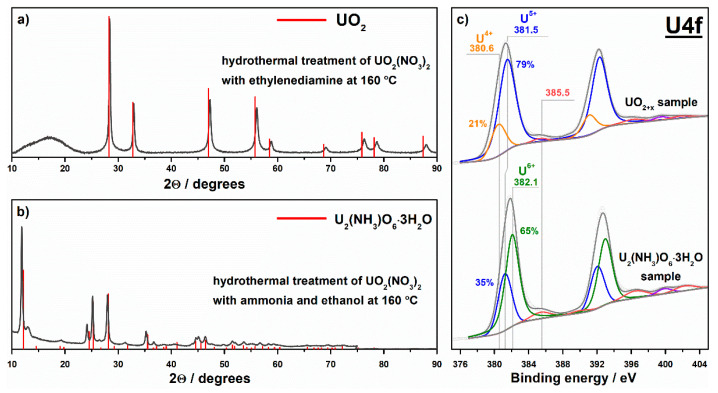
XRD patterns (**a**,**b**) and photoelectron U4f spectral regions (**c**) for the samples prepared via the hydrothermal method.

**Figure 4 nanomaterials-11-01036-f004:**
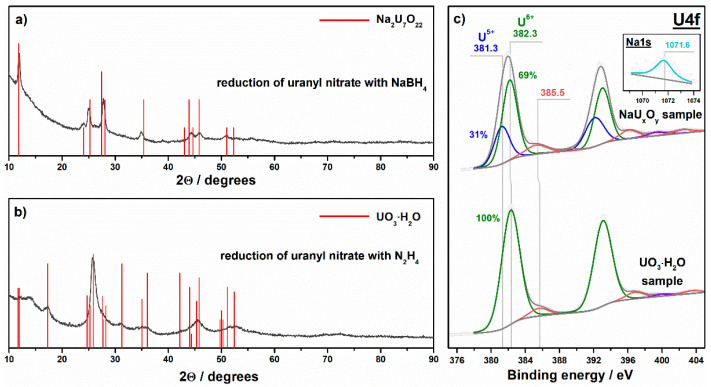
XRD patterns (**a**,**b**) and photoelectron U4f spectral regions (**c**) for the samples prepared via the chemical reduction.

**Figure 5 nanomaterials-11-01036-f005:**
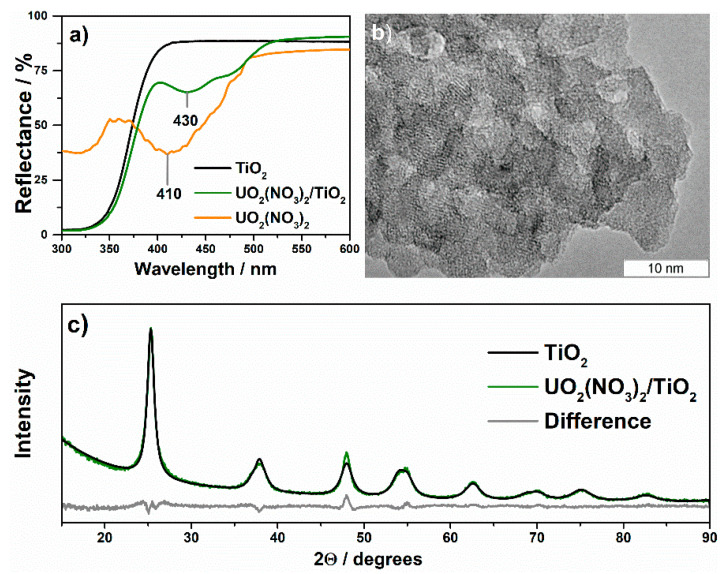
UV–vis (**a**), TEM (**b**), and XRD (**c**) data for TiO_2_ modified with uranyl nitrate.

**Figure 6 nanomaterials-11-01036-f006:**
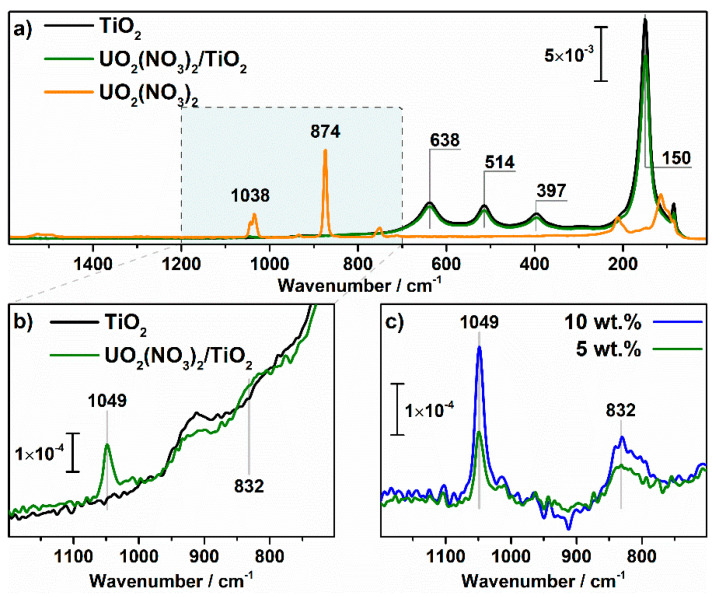
Raman spectra of UO_2_(NO_3_)_2_, TiO_2_, and UO_2_(NO_3_)_2_/TiO_2_ (**a**). (**b**,**c**) show corresponding spectra with a higher magnification in the range of 1200–700 cm^–1^.

**Figure 7 nanomaterials-11-01036-f007:**
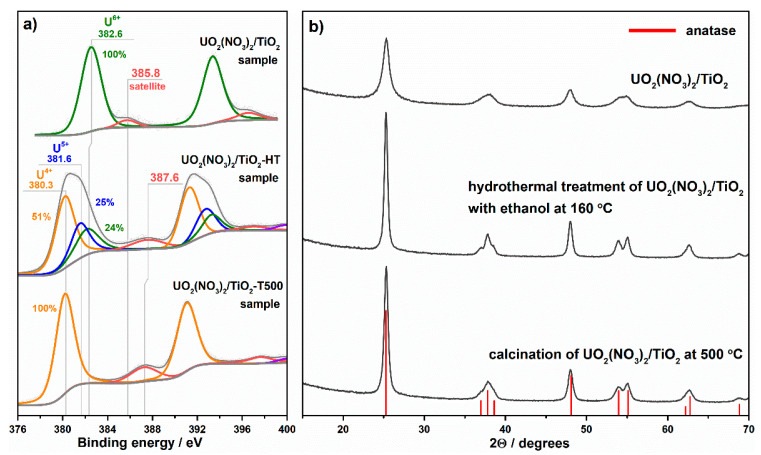
Photoelectron U4f spectral regions (**a**) and XRD patterns (**b**) for the uranium-contained TiO_2_ samples.

**Figure 8 nanomaterials-11-01036-f008:**
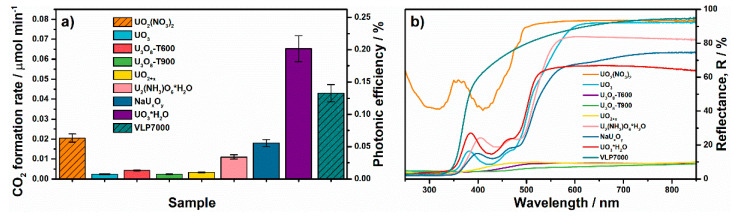
Data on photocatalytic activity (**a**) and UV–vis spectra (**b**) of solid uranium oxycompounds.

**Figure 9 nanomaterials-11-01036-f009:**
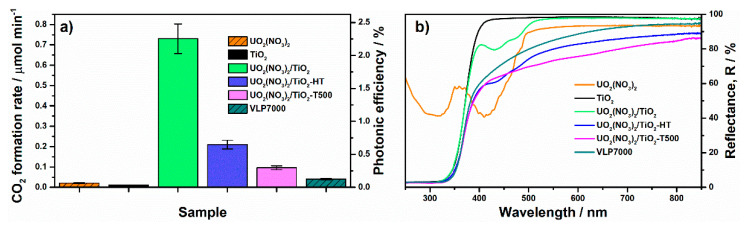
Data on photocatalytic activity (**a**) and UV–vis spectra (**b**) for TiO_2_-supported samples.

**Table 1 nanomaterials-11-01036-t001:** List of prepared uranium-contained samples and their characteristics.

Sample	Synthesis	Crystal Phase ^1^	Uranium States at Surface ^2^	a_s,BET_, m^2^ g^–1^
**UO_2_(NO_3_)_2_**	Commercial UO_2_(NO_3_)_2_·6H_2_O	UO_2_(NO_3_)_2_·6H_2_O	U^6+^ (100%) [29]	2.5 ± 0.3
**UO_3_**	Calcination of UO_2_(NO_3_)_2_·6H_2_O at 400 °C for 3 h	β-UO_3_	U^6+^ (100%)	0.6 ± 0.1
**U_3_O_8_-T600**	Calcination of UO_2_(NO_3_)_2_·6H_2_O at 600 °C for 3 h	U_3_O_8_		5.3 ± 0.1
**U_3_O_8_-T900**	Calcination of UO_2_(NO_3_)_2_·6H_2_O at 900 °C for 3 h	U_3_O_8_	U^6+^ (63%) U^4+^ (37%) [29]	1.5 ± 0.1
**UO_2+x_**	Hydrothermal treatment of uranyl nitrate in water solution with an excess of ethylenediamine at 160 °C for 72 h	U_2.12_	U^5+^ (79%) U^4+^ (21%)	<0.5 ^3^
**U_2_(NH_3_)O_6_** **·3H_2_O**	Hydrothermal treatment of uranyl nitrate in water solution with an excess of ethanol and ammonia at 160 °C for 72 h	U_2_(NH_3_)O_6_·3H_2_O	U^6+^ (65%) U^5+^ (35%)	19.1 ± 0.2
**NaU_x_O_y_**	Chemical reduction of uranyl nitrate with sodium borohydride	Na_2_U_7_O_22_	U^6+^ (69%) U^5+^ (31%)	<2.3 ^3^
**UO_3_** **·H_2_O**	Chemical reduction of uranyl nitrate with hydrazine	UO_3_·H_2_O	U^6+^ (100%)	16.4 ± 0.4
**TiO_2_**	Commercial TiO_2_ Hombifine N	Anatase		327
**UO_2_(NO_3_)_2_/** **TiO_2_**	Impregnation of TiO_2_ with water solution of uranyl nitrate followed by drying at 160 °C	n.d. ^4^/anatase	U^6+^ (100%) ^5^	298
**UO_2_(NO_3_)_2_/** **TiO_2_-HT**	Hydrothermal treatment of UO_2_(NO_3_)_2_/TiO_2_ in water suspension with an excess of ethanol at 160 °C for 48 h	n.d./anatase	U^6+^ (24%) U^5+^ (25%) U^4+^ (51%)	84
**UO_2_(NO_3_)_2_/** **TiO_2_-T500**	Calcination of UO_2_(NO_3_)_2_/TiO_2_ at 500 °C for 3 h	n.d./anatase	U^4+^ (100%)	103
**VLP7000**	Commercial TiO_2_ KRONOS^®^ vlp 7000	Anatase		250

^1^ Based on results of XRD analysis. ^2^ Based on results of XPS analysis. ^3^ The total value of sample’s surface area during measurement of N_2_ adsorption should be higher than 0.2 m^2^ for correct BET analysis; due to a small amount of sample, only an estimated value is presented. ^4^ The crystal phase for uranium compound is not detected (n.d.). ^5^ U^4+^ additionally forms under vacuum conditions and long-term exposure to X-rays.

## Data Availability

The data presented in this study are available on request from the corresponding author. The data are not publicly available due to privacy.

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
