# Peer review of "Synthesis, Characterization and Visible-Light Photocatalytic Activity of Solid and TiO2-Supported Uranium Oxycompounds"

_nanomaterials, 2021, doi:10.3390/nano11041036_

Round 1

Reviewer 1 Report

  1. The LED blue light source used in the article does not seem to be a strict visible light source that filters out ultraviolet light. A more rigorous statement should be used in the description.
  2. The sequence of transformations during the calcination of uranyl nitrate is given in Section 3.1, but some links lack strict support, such as the existence of UO2(OH)(NO3). More necessary instructions should be added.
  3. The discussion about the peak at about 17° in the XRD pattern of Figure 3a was missed.
  4. Some lines seem to be missing in Figure 9a, please correct.
  5. In Section 3.2.1, both NaUxOy and UO3H2O showed higher photocatalytic activity than UO2(NO3)2, but they were not modified with TiO2 in the following section. Please explain the reason.
  6. Uranium is widely known as a radioactive element. Are there any potential environmental hazards or health risks in the process of use?

Author Response

We would like to thank the reviewer for constructive remarks, which help us to improve the manuscript. Please find the detailed answers and comments below.

  1. According to the classifications, radiation with a wavelength from 10 to 400 nm corresponds to the ultraviolet region while radiation with a wavelength from 400 to 760-780 nm corresponds to the visible region. The LED light source used is this study emits photons with a wavelength from 405 to 520 nm, a maximum of emission is observed at 450 nm (We have added the emission spectrum of the used LED to Supplementary). According to UV-vis spectroscopy, this region corresponds to light absorption by studied materials. Therefore, taking into account the absence of UV radiation in the spectrum of this LED and the presence of a component in the visible region of optical radiation only, we refer it in the text as a "visible light source" and often use the term “visible light”. We agree that this LED does not correspond in the strict sense to the full-range visible light sources (mercury lamps, LED lamps with several types of diodes, solar light). To avoid misunderstanding, in the revised version of manuscript we have more often used the term “blue light”. Please find the revised version.
  2. The mentioned sequence of transformations during the calcination of uranyl nitrate was presented schematically based on the literature data. To avoid misunderstanding and additional discussion, we have removed the paragraph with this sequence in the revised version of manuscript because it has no significance for this study.
  3. The XRD pattern for the UO2+x sample shown in Figure 3a contains a very broad peak at ca. 17° and narrow peaks, which correspond to the crystalline phase of UO2. There are no other broad peaks in the pattern, and the observed peak at ca. 17° can be attributed to an amorphous phase. We have added this discussion to the corresponding place of text. Please find the revised version of manuscript.
  4. Figure 9a contains all lines. Maybe, it seems that bars for UO2(NO3)2, TiO2 and VLP7000 samples do not have error marks but it is only due to these samples exhibit a low activity, much lower compared to other samples presented in this figure, and these marks are very small. We have increased the resolution of this figure to improve its readability.
  5. An increase in the activity of solid uranium oxycompounds compared to the initial uranyl nitrate was observed only in the case of reduction of uranyl nitrate with hydrazine (UO3 H2O sample in this manuscript). Based on our previous experience, the treatment of uranyl-grafted TiO2 with some reducing agents, for instance, sodium borohydride or hydrazine (these methods were used in this study for the preparation of NaUxOy and UO3H2O samples), or its photoreduction in an inert atmosphere under irradiation leads to a decrease in its photocatalytic activity. These objects, which contain a low amount of uranium, are difficult for the characterization. A further strong investigation is required to make clear conclusions on the transformation that occurred during these treatments. It is out of scopes of this manuscript, which is too long even in the current state. In manuscript in the section devoted to the TiO2-supported materials, we presented the results for treatments provided a change in the chemical state of uranyl ion on the TiO2 surface (that is clearly shown as a change in the optical properties of the treated materials) to underline specific role of uranyl nitrate in the concerning system. We have added a short explanation to the text of manuscript. Please find the revised version.
  6. For the synthesis, we used uranyl nitrate precursor contained depleted uranium (U238). Yes, this isotope is also radioactive but it is a source of alfa-radiation only. Therefore, U238 is not as dangerous as other radioactive isotopes and elements because the skin will block the alpha particles. It can lead to a high impact in the case of penetration (e.g., during inhalation) into the human body due to alfa-radiation and a toxic effect. In this point of view, it is similar to other toxic elements and heavy metals (e.g., Cd). On the other hand, the viability of uranyl-grafted TiO2 as a photocatalyst may be complicated by the strict government regulations due to the presence of uranium in the composition of this photocatalyst. Therefore, we believe that the proposed method of modification has the potential in special areas of application: (i) the combination of sorption of uranium compounds from wastewaters on the surface of TiO2 or other support and associated decomposition of organic impurities under solar light during the wastewater purification; (ii) the application of uranyl-grafted TiO2in air purification systems to adjust comfort microclimate in spaceships, submarines, etc.

Reviewer 2 Report

MS No: 

nanomaterials-1171395-peer-review-v1

Title:

Synthesis, Characterization and Visible-Light Photocatalytic

Activity of Uranium Oxycompounds

Authors:     

Mikhail Lyulyukin, Tikhon Filippov, Svetlana Cherepanova, Maria Solovyeva, Igor Prosvirin, Andrey Bukhtiyarov, Denis Kozlov and Dmitry Selishchev

The present manuscript deals with the synthesis and characterization of uranium oxycompounds and discuss their photocatalytic properties under visible light irradiation. In general, the paper is well organized and written and contains very useful information concerning the physicochemical characterization and synthesis of such materials. However, the use of uranium compounds in a green technology such as photocatalysis, is completely out of practical value. In addition, the cost of such compounds is prohibitive for such applications. I believe that the paper is not suitable for publication in Nanomaterials and should be rejected.

Author Response

In our opinion, one of the purposes of public science is sharing the results and data between scientists and researchers all over the world to find the answers to basic questions and to find new areas and routes for sustainable development based on scientific knowledge. The questions on practical application, cost, and other aspects of technology run in a different way and correspond to the next level. The submitted manuscript shows new data, which can be helpful for readers, and tries to answer the question on synergistic effect in a uranyl-TiO2 system, which exhibits great results in photocatalytic oxidation of organic compounds under blue light.

Concerning the hazard aspect, the published papers describe a lot of toxic/hazard materials used in the field of photocatalysis. For instance, CdS is a toxic and cancerogenic compound but it (and other composites contained CdS in their compositions) is commonly used as a photocatalyst for the photocatalytic hydrogen evolution. It is not a reason to stop the research.

We underline in the text of manuscript possible ways for application of concerning uranyl-TiO2 system. Additionally, the combination of sorption of uranium compounds from wastewater on the surface of TiO2 or other support and associated decomposition of organic impurities under solar light can also be a possible way for its use during the purification of wastewater.

Therefore, we believe that the submitted manuscript is appropriate for publication in Nanomaterials journal because it provides new data on the photocatalytic materials and on the functionalization of TiO2 with uranyl ions for boosting its performance under visible light. The results of this study can be useful for readers for further research.

Reviewer 3 Report

This is an interesting paper concerning the preparation of uranium compounds and uranium compounds modified titanium dioxide based sorbent, followed by analysis of their photocatalytic activity towards organic pollutants (acetone was chosen as a model) and comparison with commercial TiO2 photocatalyst KRONOS® vlp 7000. The characterization of the materials is quite complete and the parameters seem to be consistent.

The advantages of the materials in comparison with traditional TiO2 materials are clearly defined: activation of the materials in visible part of UV-vis spectrum (blue light) and enhanced photocatalytic properties due to U(VI) photoreduction through different mechanisms into U(V) and U(IV). The results are not surprising and correspond with literature data for similar uranyl/TiO2 materials cited or not cited in this paper. I think that after appropriate revision the paper deserves to be published in Nanomaterials.

Specific comments

  1. Authors should discuss differences in photocatalytic properties of individual materials in more details e.g. address higher activity of materials with TiO2 towards acetone in comparison with pure uranyl compounds.
  2. The tittle needs to be revised to reflect materials with TiO2.
  3. Both description of materials preparation and characterizations are very detailed and some part/figures could be transferred to the Suplementary data, some important data evaluated from characterization techniques are summarizes in Table 1.
  4. The English could use some work in the editing process, but is basically clear enough to be read and understood. Authors should correct typing errors (ammine, exited states …)

Author Response

We would like to thank the reviewer for constructive remarks, which help us to improve the manuscript. Please find the detailed answers and comments below.

  1. According to your recommendation, we have more discussed the photocatalytic activity of different materials in the text. Please find the revised version of manuscript.
  2. The initial idea of this paper was to describe the data for solid uranium compounds only. Further, it was expanded to TiO2-supported materials. Now, we agree with your recommendation on the correction of title and suggest a new one as follows: “Synthesis, Characterization and Visible-Light Photocatalytic Activity of Solid and TiO2-Supported Uranium Oxycompounds”, to reflect both types of materials.
  3. The photocatalytic activity of uranium compound depends on its composition and the charge state of uranium. Therefore, a lot of effort in this study was spent on the synthesis of different uranium compounds. U-contained systems are difficult for characterization due to many states of uranium. Therefore, the preparation techniques and the results of important characterization methods are discussed in detail to show their correlation. Table 1 only concentrates these data. To avoid additional questions by readers on the validity of the statements made, we would like to maintain the description of results and their discussion as made before. We have tried to short the text in this section of manuscript. Please find the revised version.
  4. We have checked the text once again to correct all typing errors and some sentences. Concerning the term “ammine”, it is not typing error. In contrast to the term “amine”, which corresponds to an organic compound contained -N, -NH, or -NH2 group, the term “ammine” corresponds to NH3 ligand in the coordination sphere of complex. This term was used in the text for description of U2(NH3)O6×3H2O compound.

Round 2

Reviewer 2 Report

Accept